# Examining Bayesian network modeling in identification of dangerous driving behavior

Yichuan Peng[1,2], Leyi Cheng[2], Yuming Jiang[2]*, Shengxue Zhu[1]

**1** Jiangsu Key Laboratory of Traffic and Transportation Security, Huaiyin Institute of Technology, Huaian, China, **2** Key Laboratory of Road and Traffic Engineering, Ministry of Education College of Transportation Engineering, Tongji University, Shanghai, China

* 14191@tongji.edu.cn

## Abstract

Traffic safety problems are still very serious and human factor is the one of most important factors affecting traffic crashes. Taking Next Generation Simulation (NGSIM) data as the research object, this study defines six control indicators and uses principal component analysis and K-means++ clustering methods to get the driving style of different drivers. Then use the Bayesian Networks Toolbox (BNT) and MCMC algorithm to realize the structure learning of Bayesian network. and parameter learning was completed through Netica software. Finally, the vehicle-based traffic crash risk model was created to conduct sensitivity analysis, posterior probability inference, and simulation data was used to detect the feasibility of the model. The results show that the Bayesian network modeling can not only express the relationship between the crash risk and various driving behaviors, but also dig out the inherent relationship between different influencing factors and investigate the causes of driving risks. The results will be beneficial to accurately identify and prevent risky driving behavior.

## 1. Introduction

With the development of the transportation industry, the number of cars has increased, and the situation of road traffic safety has become more severe. The "2018 Global Road Safety Report" released by the WHO pointed out that approximately 1.35 million people die from road traffic collisions every year, 3,700 people die from car accidents every day, and one person loses his life on the road every 24 seconds. The causes leading to traffic crashes are diverse and complex [1–3], and the influencing factors mainly include human factor, vehicle factor, roads, and environmental factor [4]. According to previous research [5–11], human factor is one of the most important factors. Therefore, identification of dangerous driving behaviors in a timely manner can reduce the risk of driving and improve the safety of the road traffic system.

In 1993, the driving style was first defined as the driver's habitual driving method during driving, and it was emphasized that driving style is a unique driving attribute of each person [12]. It will affect the driver's speed control, driving awareness, driving skills and many other aspects during driving which have a great relationship with traffic safety [13–15]. Currently,

**Data Availability Statement:** Data are available from https://data.transportation.gov/Automobiles/Next-Generation-Simulation-NGSIM-Vehicle-Trajector/8ect-6jqj.

**Funding:** The authors disclosed receipt of the following financial support for the research,

authorship, and/or publication of this article: The study was jointly sponsored by the Key Projects of Soft Science of Shanghai Science and Technology Commission(no.20692111400), the Foundation for Jiangsu key laboratory of Traffic and Transportation Security (grant no. TTS2019-02), and Natural science fund for colleges and universities in Jiangsu Province (grant no. 18KJA580001). All opinions are those only of the authors.

**Competing interests:** The authors have declared that no competing interests exist.

there are two main types of driving style recognition. One is a subjective questionnaire survey. Li [16] et al. used a standard driver behavior questionnaire on 225 non-professional drivers to determine the number of driver style categories based on the fuzzy C-means (FCM) algorithm. Liu Jing [17] and others used the Multi-Dimensional Driving Style Scale and the General Decision Style Scale by surveying 199 drivers to study the relationship between various factors and driving style. However, questionnaire surveys are subjective and may affect the research results due to the driver's cognitive bias. To reduce subjectivity, this article uses another method based on vehicle kinematics parameters to identify driving style. This paper uses principal component analysis for dimensionality reduction and then uses k-means clustering analysis to classify driving style, and uses the elbow method to determine the number of classifications based on the NIGSIM data set which comes from the US "Next Generation Simulation" (NGSIM) program and will be explained in detail later.

Among the related methods of risk evaluation, the commonly used methods are fuzzy evaluation method [18], risk index method [19], regression model [20], decision tree, K-means algorithm [21], bayesian method [22, 23] and neural networks [24]. Wu et al [25] took various bad driving behaviors as evaluation indicators, and obtained the main factors affecting traffic crashes through fuzzy evaluation. Zhang [26] proposed to use equivalent acceleration as the weighting index and use the driving risk index proposed by Toledo and the safety threshold to judge whether the driving behavior is safe. Taking the accident samples as the research object, Ye [27] constructed a generalized ordered logit model to estimate the distribution probability of different severity levels of crashes and identify the main factors affecting different severity levels of rollover crashes. Sheng Dong [28] established a binary logit model to perform simulation and analysis of rear-end collisions. Zhang [29] applied the CART decision tree algorithm to focus on the driving behavior to explore the impact on the severity of the consequences of the crash. Yanyong Guo et al [30] developed traffic conflict-based real-time safety models for signalized intersections using multiple indicators under the Bayesian framework. Tarek et al [31] proposed a hierarchical Bayesian peak over threshold approach for conflict-based before-after safety evaluation of Leading Pedestrian Intervals.

In this paper, we employed Bayesian network to construct a vehicle-based traffic crash risk model. Compared with methods such as fuzzy mathematics and analytic hierarchy process, the objectivity of Bayesian network is stronger. Compared with regression models, Bayesian networks can better show the correlation between different crash risk factors in complex systems while odds ratios from a logistic regression can only show the relationship between crash risk and various factors. Compared with neural networks, Bayesian networks are more explanatory. Model visualization can directly show the relationship between various influencing factors, and make inferences by setting evidence variables. The Bayesian network graphically describes the relationships between independent and dependent variables. According to the Bayesian network structure, prior probability, and the conditional probability table of each node, the probability of event occurrence can be predicted. Therefore, the Bayesian network is selected for the prediction of traffic crash risk.

## 2. Data preparation

### 2.1 Data description

In this study, we used NIGSIM data to establish a vehicle-based risk assessment model and used the simulated data to verify the model.

NIGSIM data [32–34] comes from the US "Next Generation Simulation" program which collected vehicle trajectory data on us-101 southbound and Lankershim Avenue in Los Angeles, California, I-80 eastbound in Emeryville, California, and Peachtree Street in Atlanta,

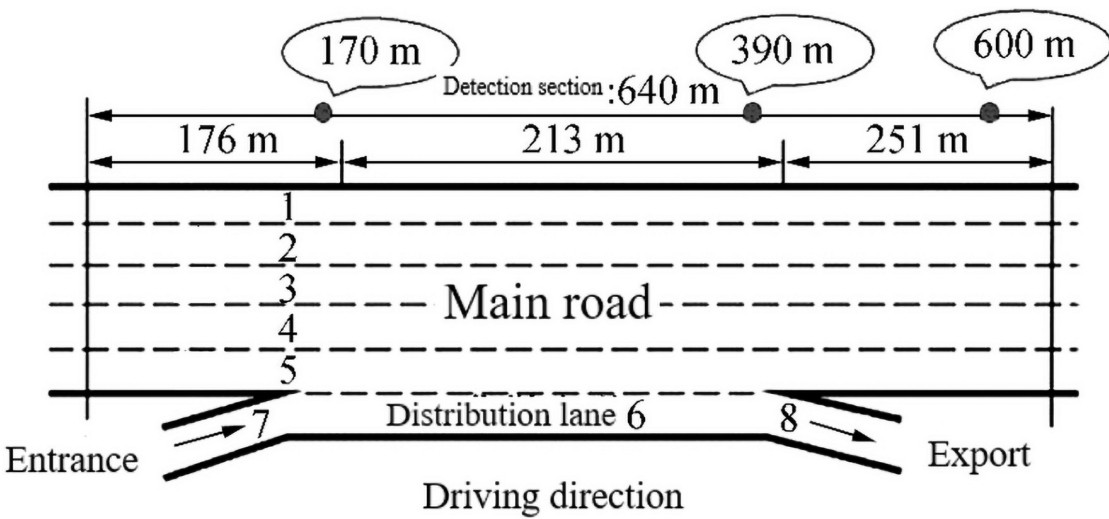

**Fig 1. The road section of us-101.**

Georgia. And this study selected us-101 data for analysis. The simulation data were collected from a driving simulation system, which can simulate the driving environment and driving behavior of the vehicle. The driver controls the vehicle model through acquisition modules such as the keyboard and steering wheel, and adopts different response methods in different driving environments.

The us-101 data contains 25 attributes such as Vehicle ID, Frame ID, Global time, Local X, and Local Y, etc. The length of the study area is 640m, including 5 main lanes; 1 distribution lane is located between the entrance of Ventura Boulevard and the exit of Cahuenga Boulevard. Fig 1 shows the road section of us-101.

We selected the vehicles on the main lane in order to ensure the accuracy of the data, which accounts for about 97% and selected the car as the research object and then converted the British unit into the international standard unit.

The contents of the data after processing is shown in Table 1:

**Table 1. Basic information of NIGSIM data.**

| number | name | unit |
|---|---|---|
| 1 | Vehicle ID | number |
| 2 | Frame ID | 100ms |
| 3 | Total frames | 100ms |
| 4 | Global time | h |
| 5 | Local X | m |
| 6 | Local Y | m |
| 7 | Vehicle length | m |
| 8 | Vehicle width | m |
| 9 | Vehicle velocity | km/h |
| 10 | Vehicle acceleration | $m/s^2$ |
| 11 | Lane Identification | number |
| 12 | Space headway | m |
| 13 | Time headway | s |

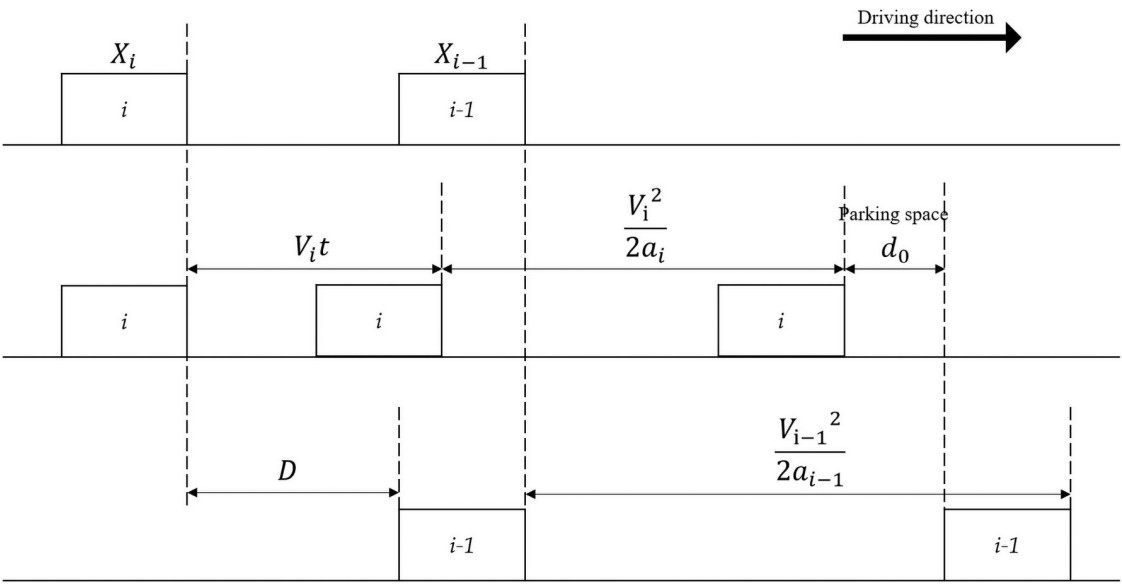

**Fig 2. Vehicle braking process.**

### 2.2 Risk assessment indicators

There are many factors influencing road traffic safety. As we mentioned at the beginning of the study, dangerous driving behavior is one of the leading causes of the traffic crashes. This part focuses on the identification of dangerous driving behaviors by establishing eight vehicle-based risk assessment indicators based on the collected data.

**2.2.1 Car following interval control indicator.**   In order to have enough reaction time to deal with unexpected accidents for the driver, a reasonable safety distance between vehicles needs to be guaranteed. The minimum distance between vehicles can be derived by analyzing the braking process of vehicles as shown in Fig 2. If the distance between front and rear vehicles is less than the minimum vehicle distance during driving, it is considered to have a certain driving risk.

$$D = V_i t + \frac{V_i^2}{2a_i} - \frac{V_{i-1}^2}{2a_{i-1}} + d_0 \qquad (2.1)$$

In the picture, $X_{i-1}$, $X_i$ means the location of the two vehicles before braking and deceleration, D represents the distance between the two vehicles, t represents the reflected time [35] of the following car which means the time from when the vehicle in front slows down to when the rear vehicle slows down, and the distance traveled by the rear vehicle during this period is reflected distance which is represented by $V_i t$. According to the dynamic formula, the braking distance of the front vehicle is $V_{i-1}^2/2a_{i-1}$, and the driving distance of the rear vehicle during braking is $V_i^2/2a_i$. Assuming that the speed and acceleration of the front and rear cars are the same before braking, the minimum vehicle spacing $D_{min}$ between them can be obtained.

$$D_{min} = V_i t + d_0 \qquad (2.2)$$

Using the ratio $\zeta$ of the minimum vehicle distance to the actual vehicle distance as the following interval control index to evaluate the driver's control of the vehicle distance, the

calculation formula of $\zeta$ is as follows:

$$\zeta = \frac{D_i}{D} \tag{2.3}$$

**2.2.2 Sharp acceleration and deceleration control indicator.** The sudden acceleration and deceleration caused by poor control will bring crash risks. Set the abrupt acceleration threshold $a'_{acc}(a'_{acc} > 0)$, and abrupt deceleration threshold $a'_{dec}(a'_{dec} < 0)$, and when abrupt acceleration is greater than $a'_{acc}$ or the abrupt deceleration is less than $a'_{dec}$, it is judged that the abrupt speed change occurs. Define the continuous rapid acceleration/deceleration time threshold as T. According to the three different states of no speeding behavior, speeding behavior, and continuous speeding behavior, the risk levels are divided into three categories: low risk, normal risk, and high risk. Compared with the previous simple classification of whether there are over-speeding behaviors based on only acceleration, this classification is more detailed and accurate.

**2.2.3 Frequent acceleration and deceleration control indicator.** Frequent acceleration and deceleration mainly mean that the speed of the vehicle changes frequently with a short period. Frequent acceleration and deceleration are not illegal in traffic laws. Therefore, drivers pay relatively little attention to such dangerous driving behaviors. What's more, in the process of frequent acceleration and deceleration, fuel consumption will increase, causing environmental pollution and economic waste. Therefore, the driver should be reminded in time when the vehicle speed is detected to be unstable.

Define Q as the oscillation frequency of acceleration in time T, n means the number of acceleration changes in the time T, so the calculation formula is:

$$Q = \frac{n}{T} \tag{2.4}$$

**2.2.4 Line driving control indicator.** Define D1 and D2 as the distance respectively between the vehicle and the left or right sides of the lane. Set the safety distance as D. There is a high driving crash risk when the condition D1<D/2 or D2< D/2 is satisfied. Considering that the data with a small value may be caused by two reasons: driving on the line or changing lanes, a time threshold T is set to eliminate the interference of the lane changing behavior.

**2.2.5 Serpentine driving control indicator.** The serpentine driving control indicator is created to study the dangerous driving situation where the vehicle shakes frequently in a short period of time based on the distance collected by the vehicle from the left side of the lane every time.

Within time T, the sloshing frequency W of the vehicle is determined by judging the change of the distance of the vehicle relative to the left side of the lane. When the value of W is larger, the driving behavior is more dangerous. The calculation formula of W is as 2.5:

$$W = \frac{k}{T} \tag{2.5}$$

Where k is the number of shaking in time T.

**2.2.6 Speeding control indicator.** Set a speed threshold V, and when the vehicle speed exceeds the threshold V, it is judged that an overspeed behavior has occurred. Then for each overspeed behavior, calculate the overspeed duration, and define the overspeed time threshold T.

**Table 2. Driving style evaluation index.**

| num | indicators |
| --- | --- |
| 1 | Average speed |
| 2 | Standard deviation of speed |
| 3 | Mean forward acceleration |
| 4 | Standard deviation of forward acceleration |
| 5 | Mean value of negative acceleration |
| 6 | Standard deviation of negative acceleration |
| 7 | Mean value of absolute acceleration |
| 8 | Standard deviation of absolute acceleration |
| 9 | Average headway distance |
| 10 | Standard deviation of headway |
| 11 | Mean value of absolute Acceleration shock |
| 12 | Standard deviation of absolute Acceleration shock |
| 13 | Number of lane changes |

**2.2.7 Frequent lane change control indicator.** Define P as the frequency of lane change in T time, the calculation formula is:

$$P = \frac{n}{T} \tag{2.6}$$

N is the number of lane changes in T time, and the value of N increases by 1 when the number of lanes where the vehicle is located changes.

**2.2.8 Driving style.** Driving style is closely related to driving safety and aggressive driving style is usually more likely to cause traffic crashes. In order to combine with the actual situation and reduce the impact of subjective questionnaire surveys on the results, this paper used vehicle kinematics data to classify driving styles.

Speed and acceleration can show the driving habits, and the frequency of lane changes and the following distance can reflect the driving personality. This article selected 13 evaluation indicators about driving style, as shown in Table 2:

Then use the principal component analysis method to reduce the dimension, and finally use the elbow method to determine the number of driving style classifications. The calculation formula of the core index SSE (sum of the squared errors) is as follows:

$$\text{SSE} = \sum_{i=1}^{k} \sum_{p \in C_i} |p - m_i|^2 \tag{2.7}$$

Among them, $C_i$ is the i_th cluster, p is the sample point in $C_i$, $m_i$ is the centroid of $C_i$ (mean of all samples in $C_i$), and SSE is the clustering error of all sample, representing the quality of the clustering effect.

## 3. Methods

Bayesian Networks graphically describe the independent or dependent relationship between variables. According to the Bayesian network structure, prior probabilities and the conditional probability table of each node, the probability of event occurrence can be predicted, which can intuitively show the causal relationship between data variables. This method is suitable for describing the relative relationship between multiple variables in a complex system. Suppose A and B are two random variables, A = a is a certain hypothesis, and B = b is a set of evidence.

Before considering the evidence B = b, the probability estimation P(A = a) of the event A = a is called the prior probability; after considering the evidence B = b, the probability estimation P (A = a) of the event A = a is called the posterior probability. Bayes' theorem describes the relationship between the prior probability and the posterior probability. The formula is as follows:

$$P(A = a \mid B = b) = \frac{P(A = a) * P(B = b \mid A = a)}{P(B = b)} \qquad (2.8)$$

The construction of Bayesian network mainly has two processes: network structure and parameter learning. Structural learning mainly includes expert experience and machine learning methods. Compared with the expert experience method, the machine learning method can avoid the influence of subjective factors. Machine learning methods include scoring-based search methods, constraint-based methods, and random sampling-based methods. The basic idea of the score-based search method is to traverse all possible structures, and then use a certain standard to measure each structure to find the best structure. In 1992, Cooper and Herskovits proposed the first Bayesian scoring function, the K2 scoring function [36, 37]; in 1995, Heckerman proposed the BD scoring function, which is a generalization of the K2 function; at the same time, Heckerman proposed the BDe scoring function based on additional likelihood equivalence hypothesis [38, 39]; Bouckaert and Suzuki proposed the K3 algorithm using a scoring function based on the Minimum Description Length (MDL) principle in information theory [40, 41]. Constraint-based Bayesian network structure learning method (also known as dependency analysis method or conditional independence test method), usually uses statistical or information theory methods to quantitatively analyze the dependence relationship between variables to obtain the optimal expression of the network structure. In 1993, the SGS algorithm proposed by Spines et al. was a typical algorithm for determining the topological structure by conditional independence tests [42]; in 2000, Spines et al. enhanced the SGS algorithm and proposed the PC algorithm [43]; in 2002, Cheng combined information theory with statistical testing and proposed the TPDA algorithm [44].

Among the machine learning methods, the learning method based on score search has a large search space and low learning efficiency and it is difficult for the method based on constraints to judge the independence between nodes. Therefore, this paper uses MCMC to learn parameters based on random sampling, which has high learning efficiency and is easy to implement. Since there is no missing data in this article, in order to improve the efficiency of parameter learning, this article used a counting algorithm for parameter learning.

In MCMC, the likelihood function is given as

$$E[f(x)] \approx \frac{1}{m} \sum_{i=1}^{m} f(x_i).(x_0, x_1, \ldots, x_m) \sim MC(p) \qquad (2.9)$$

Among them, $x_i$ represents the i-th sampling sample, m represents the number of samples and $MC(p)$ stands for Markov process.

And for counting algorithm, before it begins, the net starts off in a state of ignorance. At each node, all CPT probabilities start as uniform. Only nodes for which the case supplies values for all of its parents, have their experience and conditional probabilities modified. Each of these nodes is modified as follows.

Only the single experience number, and the single probability vector, for the parent configuration which is consistent with the case is modified. The new experience number (exper') is

found from the old (exper) by:

$$exper^{'} = exper + degree \qquad (2.10)$$

where degree is the multiplicity of the case.

Within the probability vector, the probability for the node state that is consistent with the case is changed from probc to probc' as follows:

$$probc^{'} = (probc * exper + degree)/exper^{'} \qquad (2.11)$$

The other probabilities in that vector are changed by:

$$probc^{'} = (probc * exper)/exper^{'} \qquad (2.12)$$

where probc is the probability of the case.

## 4. Results

In the driving style recognition section, we calculated the contribution rate and cumulative contribution rate of each principal component, as shown in Fig 3. According to the principle that the cumulative contribution rate reaches 85%, the first six principal components were selected to reflect the information of the original indicators sufficiently. The principal component coefficient matrix is shown in Table 3.

The score of the principal component according to the analysis coefficient will be used as the input for the classification and driving style recognition model later.

Then use the elbow method to determine the number of categories. When the number of clusters increases, the degree of aggregation of each cluster will also increase, and the SSE will gradually decrease. When the value of k is less than the correct number of clusters, the increase of the value of K will significantly increase the degree of aggregation of each cluster, and the decrease of SSE is larger. Conversely, when k reaches the true number of clusters, the return on the degree of polymerization obtained by increasing k will quickly decrease, so the decline

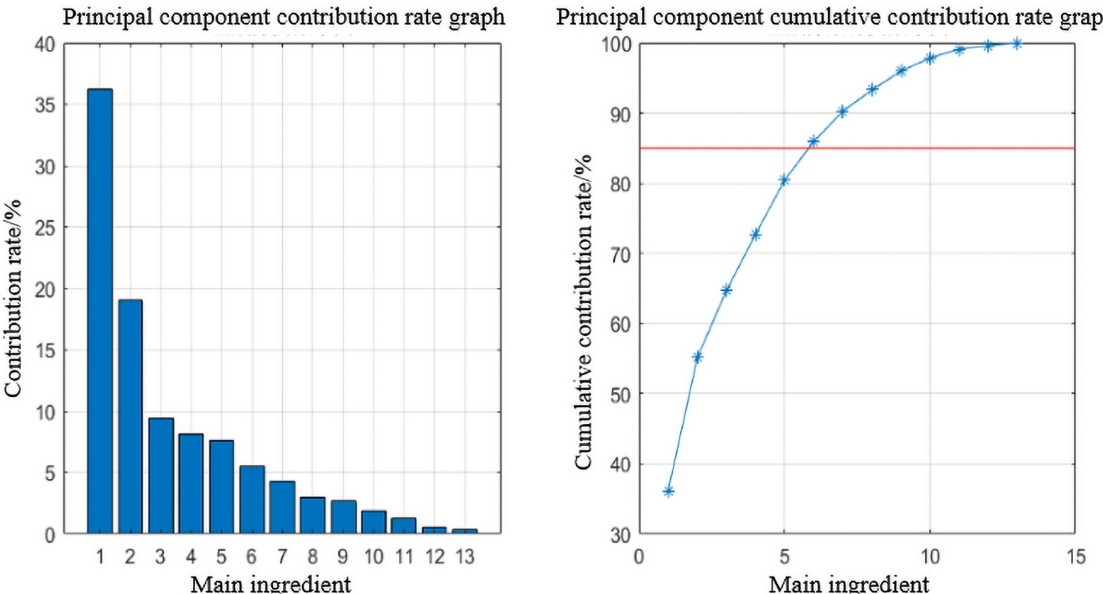

**Fig 3. Principal component contribution rate and cumulative contribution rate.**

**Table 3. Principal component score coefficient matrix.**

| Standardized variable | t1 | t2 | t3 | t4 | t5 | t6 |
|---|---|---|---|---|---|---|
| 'Mean speedX1' | 0.2548 | -0.3843 | -0.0813 | 0.0580 | 0.0093 | 0.3162 |
| 'Standard deviation of speedX2' | -0.1047 | 0.4000 | -0.0987 | 0.1552 | 0.1628 | 0.7174 |
| 'Mean forward accelerationX3' | 0.3200 | 0.1620 | -0.4574 | 0.0040 | -0.0514 | 0.1270 |
| 'Standard deviation of forward accelerationX4' | 0.2131 | 0.2764 | -0.5084 | 0.1277 | -0.1655 | -0.4131 |
| 'Mean value of negative accelerationX5' | -0.3028 | -0.1581 | -0.4851 | -0.0890 | 0.1072 | -0.0524 |
| 'Standard deviation of negative accelerationX6' | 0.2238 | 0.2667 | 0.5197 | 0.1835 | -0.2187 | -0.1168 |
| ' Mean value of absolute acceleration X7' | 0.4188 | 0.0272 | -0.0232 | 0.0310 | 0.0169 | 0.2776 |
| 'Standard deviation of absolute acceleration X8' | 0.3871 | 0.2703 | -0.0410 | 0.1598 | -0.1895 | -0.1059 |
| ' Average headway distance X9' | -0.1945 | 0.4094 | 0.0299 | -0.3083 | 0.1562 | -0.1161 |
| ' Standard deviation of headway X10' | -0.1844 | 0.4960 | 0.0209 | -0.0759 | 0.2263 | 0.0182 |
| ' Mean value of absolute Acceleration shock X11' | 0.3821 | -0.0554 | 0.0387 | -0.3230 | 0.3477 | 0.0391 |
| ' Standard deviation of absolute Acceleration shock X12' | 0.3014 | -0.0032 | 0.0727 | -0.4527 | 0.4575 | -0.1731 |
| 'Number of lane changes X13' | 0.0233 | -0.0554 | 0.0171 | 0.6914 | 0.6649 | -0.2206 |

of SSE will also decrease, and eventually it will tend to be gentle. Fig 4 is the SSE change graph when the number of clusters is between 1 and 7.

As it can be seen from Fig 4, when the number of clusters is greater than 3, the change in SSE tends to be flat. Combined with current research, driving styles are mainly divided into three types: calm type, general type, and aggressive type. So, the number of driving style classifications in this study is 3.

Then draw the result of driving style recognition based on the first three principal components. The recognition result is shown in Fig 5,

This article divided the driver's driving style into three types: calm type, general type, and aggressive type according to the degree of aggressiveness. It can be seen from the figure that the three driving styles have obvious differences in the first three principal components.

According to the previous crash risk assessment indicators, all data was used as the input for constructing the Bayesian network, and the following Table 4 was obtained by discretizing the variable of each node.

In order to facilitate the display of the Bayesian network structure, the variable names were simplified. The corresponding relationship is shown in Table 5:

The MCMC algorithm was applied to learn the Bayesian structure of the original data. Finally, there are 21 directed edges, and the DAG structure is shown in Fig 6. 0 means there is no obvious dependency between the two nodes, 1 means there is an obvious correlation between the them.

Combining the existing prior knowledge, 18 directed edges were finally determined. The visualization of the Bayesian network structure is shown in the following Fig 7.

It can be found from the above figure that after structural learning, the car following interval control, rapid acceleration and deceleration, frequent acceleration and deceleration, line driving, snake driving, speeding, and lane change frequency all have a direct impact on the type of risk. At the same time, there is a mutual influence between various factors. For example, speeding will affect the car following index, the car following index is mainly related to the vehicle distance and vehicle speed and the vehicle speed obviously affects the distance between the front and rear vehicles. Overall, the Bayesian network structure conforms to basic logical cognition.

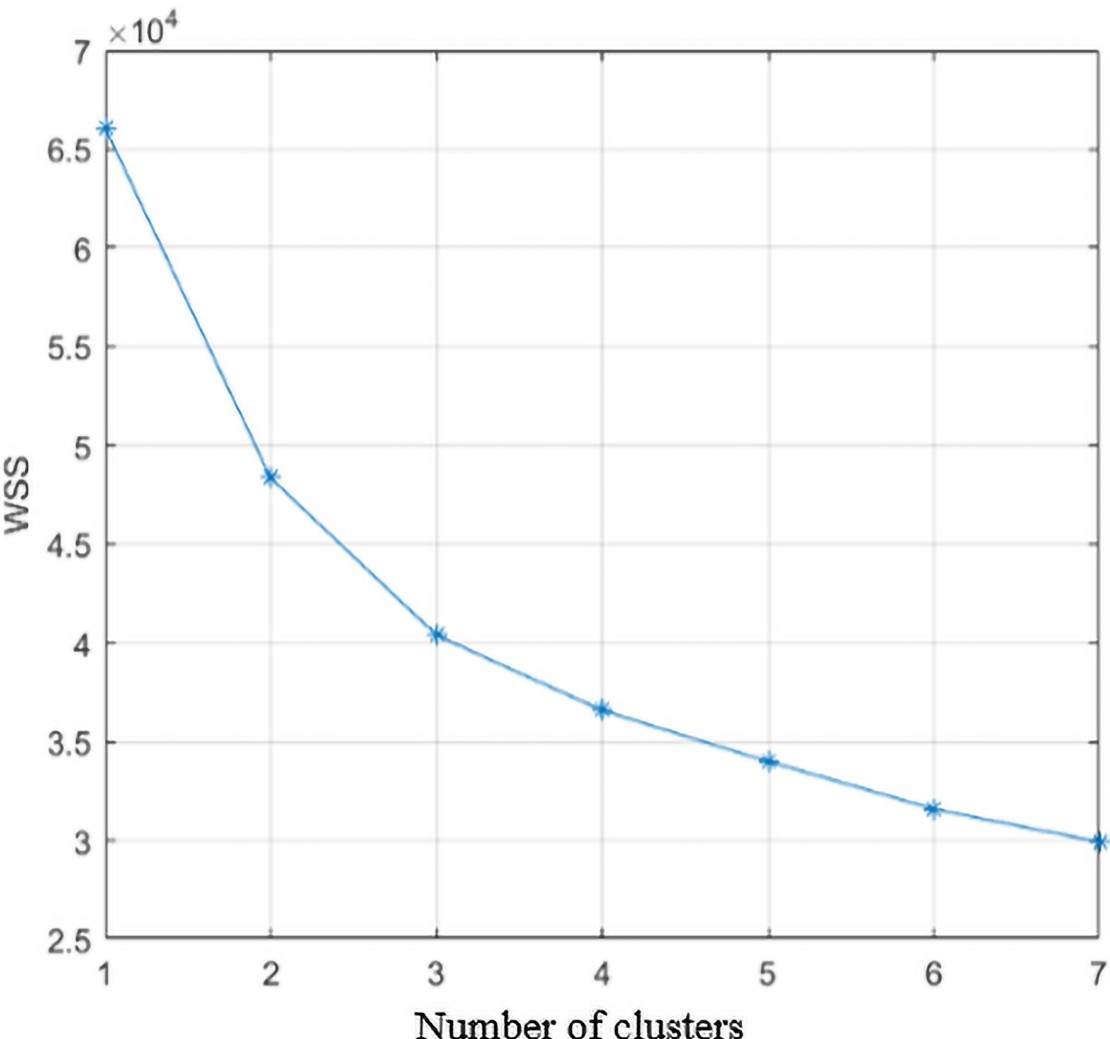

**Fig 4. Relationship between cluster number and SSE.**

The parameters were estimated based on counting-learning algorithm which is a kind of Bayesian learning algorithm. Build a Bayesian network based on the results of structural learning, and obtain the Bayesian network structure. 303 3 And then get the conditional probability table of each node by counting algorithm, the Bayesian network model obtained by parameter learning in Netica software is shown in Fig 8:

Through the counting algorithm, Netica can get the probability table of each node. The probability table shows the probability relationship between the changed node and its child nodes. Taking the Sty node as an example, this node has two child nodes Sha and Fol. Table 6 shows the conditional probability table of node Sty.

## 5. Discussion

To further explore the results of established vehicle-based crash risk model, we analyzed and verified the model from three aspects: model sensitivity, posterior probabilistic inference, and effectiveness.

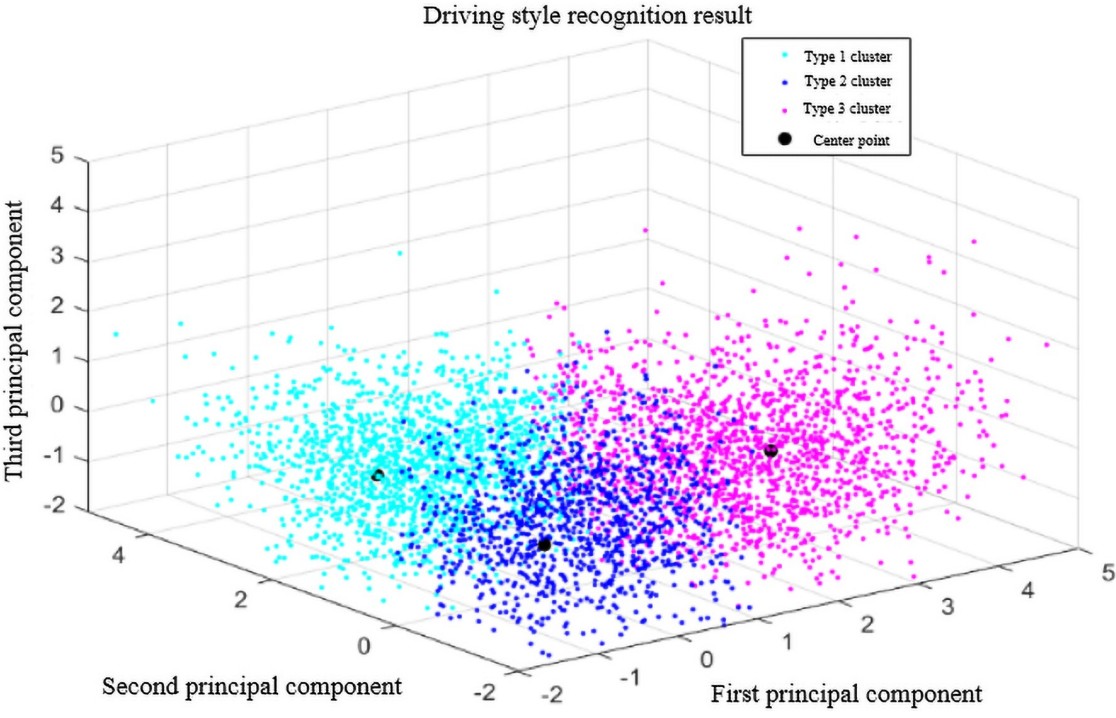

**Fig 5. Driving style recognition results.**

## 5.1 Model sensitivity

Bayesian network sensitivity analysis refers to analyzing the impact of other nodes on the target node. Through sensitivity analysis, we can identify the factors that have a greater impact on the vehicle's crash risk and take effective measures for the factors. The results are shown as follows in Table 7.

Mutual Information is used to measure the degree of dependence between nodes. The mutual information between two nodes can indicate whether the two nodes depend on each other. From Table 7, it can be seen that the mutual information between the line driving and the risk type is the largest which can be speculated that the vehicle has a greater possibility of line driving followed by snake driving, frequent acceleration and deceleration, speeding and so on.

## 5.2 Posterior probabilistic inference

The Bayesian network model can be used for probabilistic inference, including calculating the posterior probability of the target node, predicting the possibility of the result, and analyzing the main influencing factors of the result when the node state has been determined. The above two posterior probabilities of inferring results from causes and inferring causes from results are called risk prediction and causal inference, respectively.

Risk prediction refers to inputting the determined status of node variables into the Bayesian network. In Netica, if a certain node variable is determined, the corresponding state will be set to 100%, and the changes of other nodes in the entire Bayesian network can be observed. As shown in Fig 9, when the rapid acceleration and deceleration indicator is at the general risk level, the existing risk increases from 38.5% to 57.7%, and the risk level increases from a certain risk to a higher risk.

**Table 4. Bayesian network node variables and their discrete values.**

| node variables | Data description | Discrete value | frequency | ratio |
|---|---|---|---|---|
| Car following interval control indicator | $\zeta \geqq 3.5$ | 3 | 240 | 0.05% |
| | $1 \leqq \zeta < 3.5$ | 2 | 31728 | 6.80% |
| | $\zeta < 1$ | 1 | 434524 | 93.15% |
| Sharp acceleration/deceleration control indicator | $|a| \geqq 3$ & $t \geqq 3$ | 3 | 111 | 0.03% |
| | $|a| \geqq 3$ & $t < 3$ | 2 | 39815 | 8.53% |
| | $|a| < 3$ | 1 | 426566 | 91.44% |
| Frequent acceleration and deceleration control indicator | $Q \geqq 0.8$ | 3 | 21320 | 4.57% |
| | $0.6 \leqq Q < 0.8$ | 2 | 53812 | 11.54% |
| | $Q < 0.6$ | 1 | 391360 | 83.89% |
| Line driving control indicator | $(D1 < 0.25 \mid D2 < 0.25)$ & $t \geqq 5$ | 3 | 116203 | 24.91% |
| | $(D1 > 0.25$ & $D2 > 0.25) \mid ((D1 < 0.25 \mid D2 < 0.25)$ & $t < 5)$ | 1 | 350289 | 75.09% |
| Serpentine driving control indicator | $W \geqq 0.45$ | 3 | 29229 | 6.27% |
| | $0.22 \leqq W < 0.45$ | 2 | 64147 | 13.75% |
| | $W < 0.22$ | 1 | 373116 | 79.98% |
| Speeding control indicator | $V > = 55$ & $t > = 10$ | 3 | 3413 | 0.73% |
| | $V > = 55$ & $t < 10$ | 2 | 24042 | 5.15% |
| | $V < 55$ | 1 | 439037 | 94.11% |
| Frequent lane change control indicators | $P > = 0.14$ | 3 | 952 | 0.20% |
| | $P < 0.14$ | 1 | 465540 | 99.80% |
| Driving style | Aggressive type | 3 | 106063 | 22.74% |
| | General type | 2 | 144126 | 30.90% |
| | Calm type | 1 | 216303 | 46.37% |
| Risk type | High risk | 2 | 181008 | 38.80% |
| | Low risk | 1 | 285484 | 61.20% |

Another advantage of Bayesian network is causal inference. Causal inference refers to two-way inference through Bayesian network. It can not only calculate the probability of the target node, but also calculate the posterior probability of other nodes when the target node is determined. To find out the most probable combination of factors, this analysis is more intuitive and can prevent the most influential factors in advance.

Assuming that the risk probability is 100%, as shown in Fig 10, it can be discovered that the safety of line driving and serpentine driving are significantly reduced. The most obvious changes in indicators are: the range of high risk increases from the initial 24.9% to 64.7%. This shows that in the absence of other evidence, the most likely cause is line driving.

**Table 5. Node variable symbol correspondence.**

| number | variable name | Simplified symbol |
|---|---|---|
| 0 | Risk type | Typ |
| 1 | Car following interval control indicator | Fol |
| 2 | Sharp acceleration/deceleration control indicator | Sha |
| 3 | Frequent acceleration and deceleration control indicator | Qui |
| 4 | Line driving control indicator | Lin |
| 5 | Serpentine driving control indicator | Sna |
| 6 | Speeding control indicator | Ove |
| 7 | Frequent lane change control indicators | Cha |
| 8 | Driving style | Sty |

|     | Typ | Fol | Sha | Qui | Lin | Sna | Ove | Cha | Sty |
|-----|-----|-----|-----|-----|-----|-----|-----|-----|-----|
| Typ | 0   | 0   | 0   | 0   | 0   | 0   | 0   | 0   | 0   |
| Fol | 1   | 0   | 0   | 0   | 0   | 0   | 0   | 1   | 1   |
| Sha | 1   | 0   | 0   | 0   | 1   | 0   | 1   | 0   | 1   |
| Qui | 1   | 0   | 0   | 0   | 0   | 0   | 0   | 1   | 0   |
| Lin | 1   | 0   | 0   | 0   | 0   | 0   | 0   | 0   | 1   |
| Sna | 1   | 0   | 0   | 1   | 0   | 0   | 0   | 0   | 0   |
| Ove | 1   | 1   | 0   | 1   | 1   | 1   | 0   | 0   | 0   |
| Cha | 1   | 0   | 0   | 0   | 0   | 0   | 0   | 0   | 0   |
| Sty | 0   | 0   | 0   | 1   | 0   | 1   | 0   | 1   | 0   |

**Fig 6. DAG structure matrix.**

### 5.3 Effectiveness

For the judgment of the effectiveness of the crash risk model, this paper selected the risky driving process and compared it with the normal driving process from simulation data.

Combine the video of the corresponding time to verify the risk status.

Fig 11 is a screenshot of the video corresponding to the normal driving time. The vehicle speed is lower and the driving is more stable, which is consistent with the driving behavior during normal driving.

Fig 12 is a screenshot of the video corresponding to the risky driving time. During this period, driving behaviors such as frequent vehicle lane change and continuous overtaking occur, which are consistent with the driving behavior when there is an accident risk.

Then input the data into the Bayesian network to calculate the risk value, and draw a comparison chart of the risk value during the normal driving period and the risky period in Fig 13 as follows:

It can be seen from Fig 13 that during the operation of the vehicle, the risk state will change with time. The risk value in the risk driving process shown in the figure above fluctuates

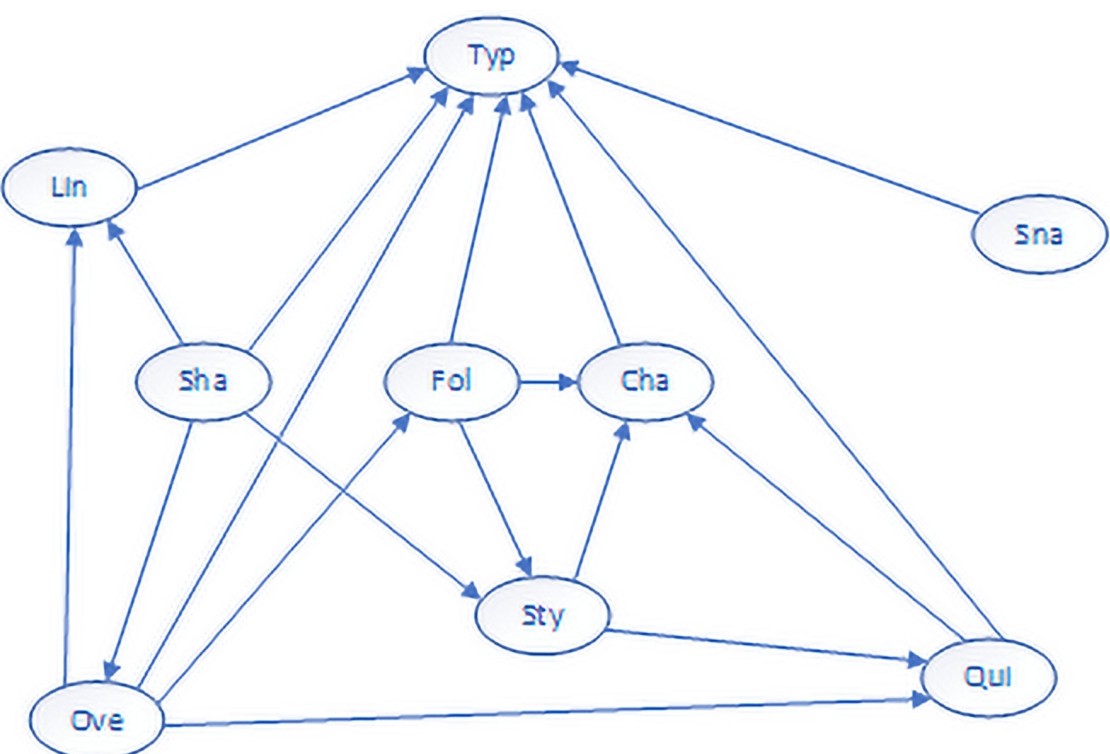

**Fig 7. Bayesian network structure diagram.**

between 50-100%. For the first 0.5s, the risk value is close to 90% which can be considered that there is a higher risk in this time period. After 0.5s, as the driver correcting the behavior, the risk is reduced and remains at about 50%. The risk of normal driving process fluctuates around 10%, which is much lower than the former one, indicating that the risk is relatively low during normal driving.

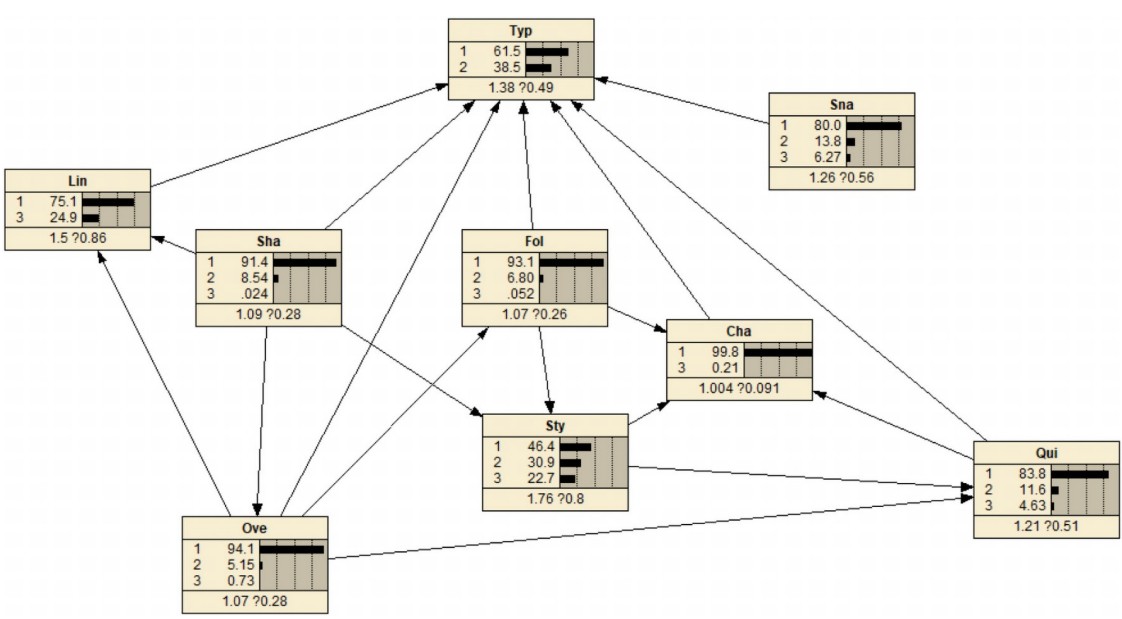

**Fig 8. Bayesian network model after learning with Netica software parameters.**

**Table 6. CPT of node Sty.**

| Sha | Fol | 1 | 2 | 3 |
|---|---|---|---|---|
| 1 | 1 | 48.038 | 31.628 | 20.334 |
| 1 | 2 | 24.767 | 36.346 | 38.887 |
| 1 | 3 | 50 | 26.852 | 23.148 |
| 2 | 1 | 46.994 | 19.645 | 33.361 |
| 2 | 2 | 23.932 | 21.176 | 54.892 |
| 2 | 3 | 50 | 10 | 40 |
| 3 | 1 | 46.602 | 10.68 | 42.718 |
| 3 | 2 | 14.286 | 7.143 | 78.571 |
| 3 | 3 | 33.333 | 33.333 | 33.333 |

**Table 7. Sensitivity analysis results of node Typ.**

| Node | Mutual Info | Percent |
|---|---|---|
| Typ | 0.96122 | 100 |
| Lin | 0.44593 | 52.9 |
| Sna | 0.10001 | 12 |
| Qui | 0.07685 | 9.38 |
| Ove | 0.02205 | 2.96 |
| Sha | 0.01035 | 1.48 |
| Fol | 0.00977 | 1.40 |
| Cha | 0.00191 | 0.252 |
| Sty | 0.00092 | 0.128 |

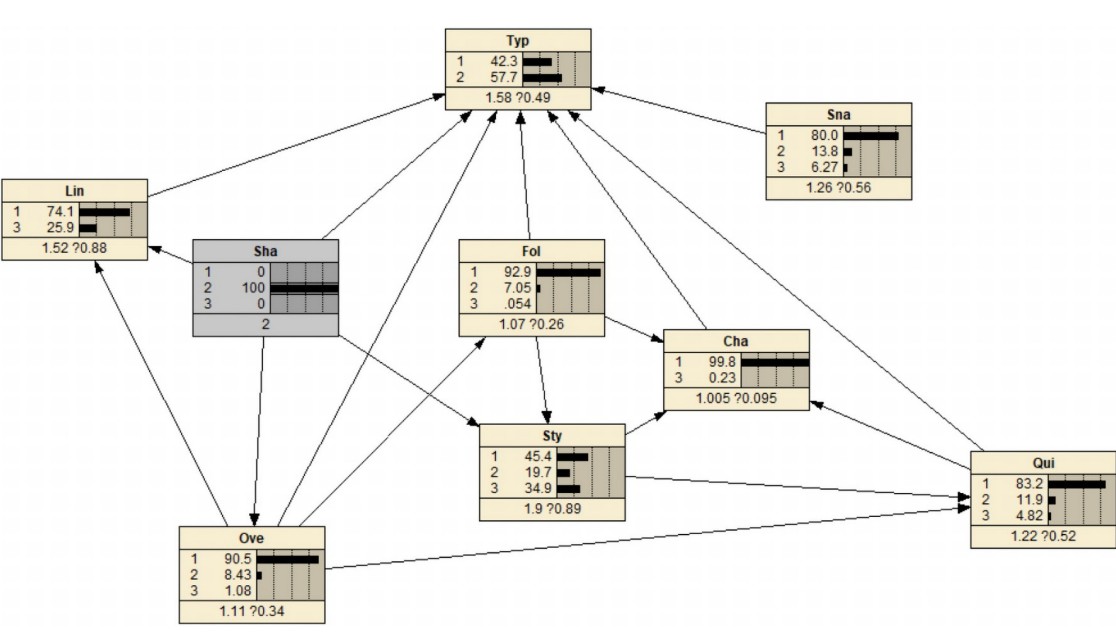

**Fig 9. Known network changes on rapid acceleration and deceleration.**

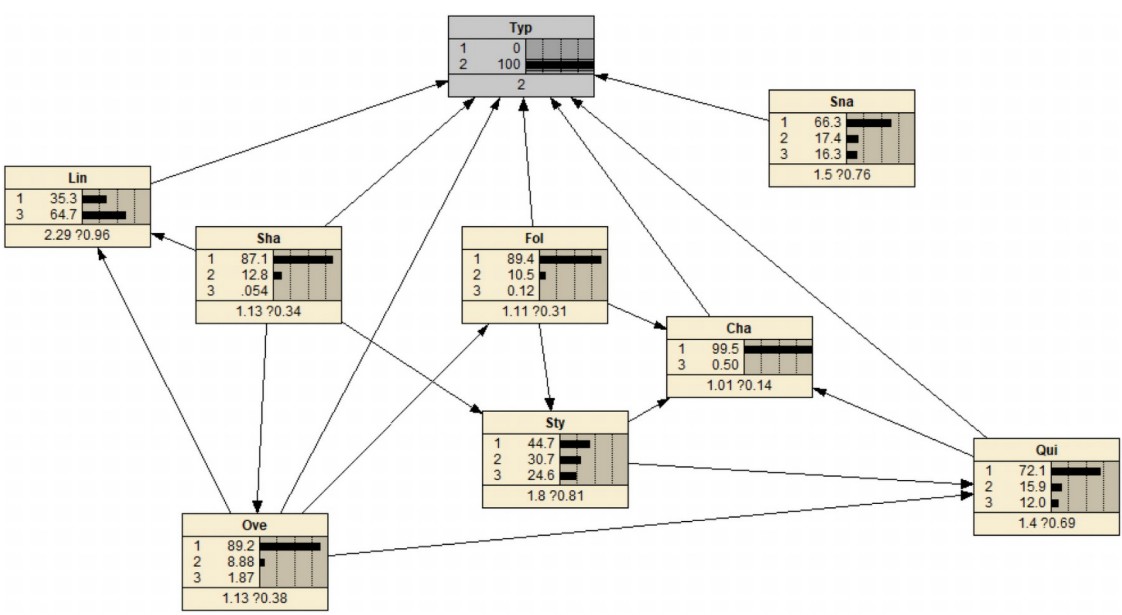

**Fig 10. Network changes with known risk status.**

Verification results show that the model is effective for vehicle-based crash risk analysis.

## 6. Conclusion

Based on the NIGSIM data set, this paper has developed 8 indicators to comprehensively identify dangerous driving behaviors. Compared with previous studies, this paper considered

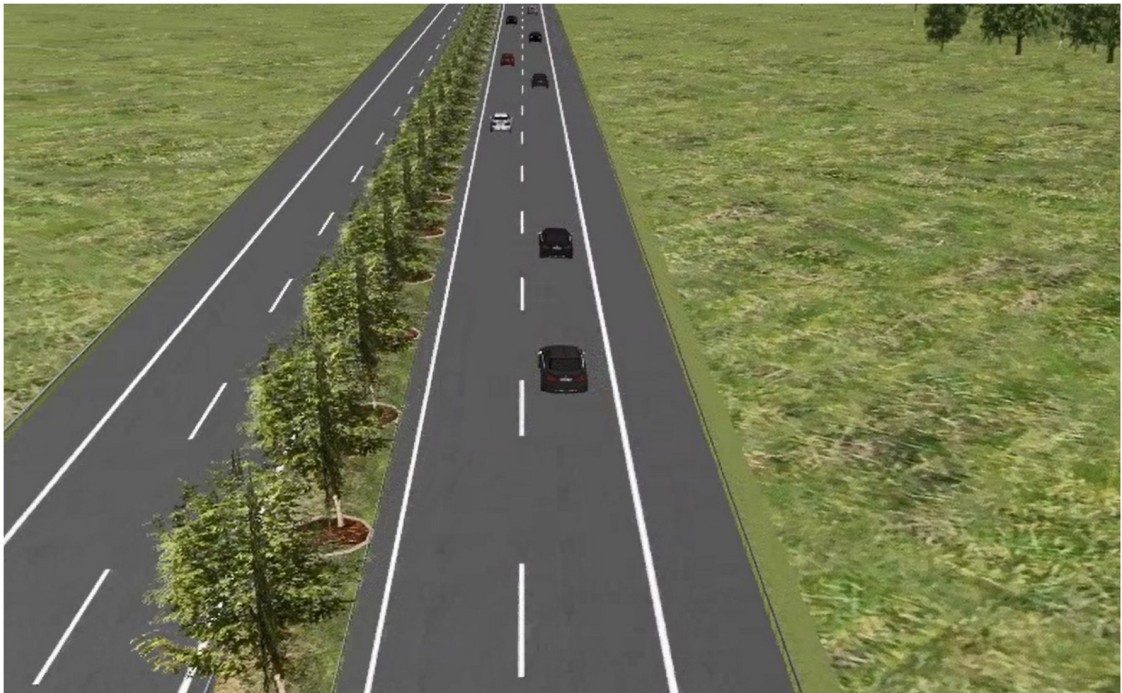

**Fig 11. Video during normal driving time.**

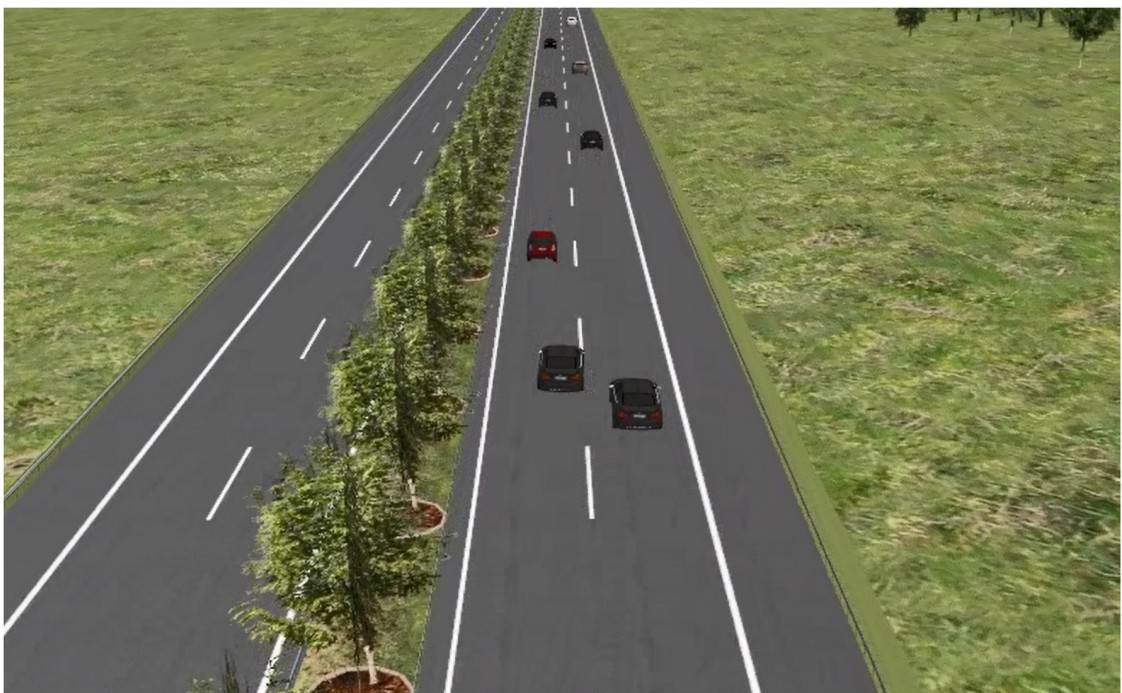

**Fig 12. Video of the time of risk driving.**

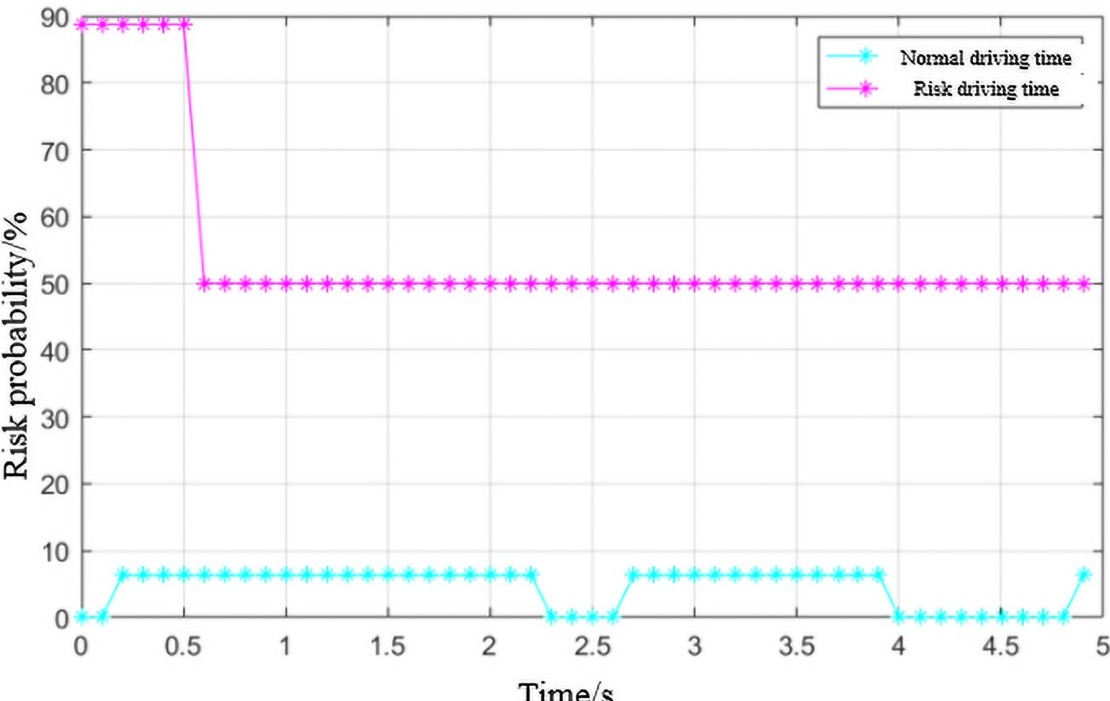

**Fig 13. Vehicle operation risks in different time periods.**

more types of dangerous driving behaviors, which can ensure the safety of vehicle operation better.

Based on the vehicle kinematics data, PCA algorithm was used for dimensionality reduction, and improved K-means algorithm was used for driving style classification. Compared with commonly used questionnaire survey methods, this method has better feasibility and objectivity.

This paper proposes to use Bayesian network to build a vehicle operation risk assessment model. Compared with commonly used methods such as fuzzy mathematics and neural network, Bayesian network is more objective and explanatory, and can analyze the correlation between various factors in depth. Through sensitivity analysis of eight factors, the mutual information between driving on the line and risk types was the largest, reaching 44.59%. It is more likely for vehicles to run on the line, which has the greatest impact on the risk types and is the most sensitive. Through causal inference, when the risk probability is 100%, the safety of line pressing and snake driving is significantly reduced. Among them, the driving index of pressing line is in a high risky range, which increases from the initial 24.9% to 64.7%. This shows that, in the absence of other evidence, the most likely cause of the risk is driving on the line, which is consistent with the conclusion of sensitivity analysis, thereby providing a research basis for the prevention of dangerous driving behavior. Once the risk level reaches a higher level, the model infers the most likely driving behavior that causes the danger based on the posterior probability, and reminds the driver to respond. Finally, the validity of the model is tested to verify that the model is effective for vehicle operation risk analysis, so that it provides a direction for the prevention of dangerous driving behaviors to fundamentally reduce the driving risk. This article only studied the risk of single-vehicle crashes, and did not comprehensively consider the interaction between other motor vehicles, which could be considered in the subsequent research.

## Author Contributions

**Data curation:** Leyi Cheng, Shengxue Zhu.

**Formal analysis:** Yichuan Peng, Leyi Cheng.

**Funding acquisition:** Yichuan Peng, Shengxue Zhu.

**Methodology:** Leyi Cheng.

**Resources:** Yichuan Peng.

**Supervision:** Yichuan Peng, Yuming Jiang, Shengxue Zhu.

**Writing – original draft:** Leyi Cheng.

**Writing – review & editing:** Yichuan Peng, Leyi Cheng, Yuming Jiang, Shengxue Zhu.

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
