## [Decision Letter · Decision Letter 0]

16 Feb 2021

PONE-D-21-00035

Examining Bayesian network modeling in identification of dangerous driving behavior

PLOS ONE

Dear Dr. Cheng,

Thank you for submitting your manuscript to PLOS ONE. After careful consideration, we feel that it has merit but does not fully meet PLOS ONE’s publication criteria as it currently stands. Therefore, we invite you to submit a revised version of the manuscript that addresses the points raised during the review process.

We look forward to receiving your revised manuscript.

Kind regards,

Yanyong Guo, Ph.D

Academic Editor

PLOS ONE

Journal Requirements:

2.We suggest you thoroughly copyedit your manuscript for language usage, spelling, and grammar. If you do not know anyone who can help you do this, you may wish to consider employing a professional scientific editing service.  

"This research has been supported by project no.20692111400."

5. Please ensure that you refer to Figure 2.2, 2.6,  2.7 and  4.4in your text as, if accepted, production will need this reference to link the reader to the figure.

6. We note you have included a table to which you do not refer in the text of your manuscript. Please ensure that you refer to Table 3.2 in your text; if accepted, production will need this reference to link the reader to the Table.

Reviewers' comments:

Reviewer's Responses to Questions

**Comments to the Author**

1. Is the manuscript technically sound, and do the data support the conclusions?

Reviewer #1: Partly

Reviewer #2: Yes

2. Has the statistical analysis been performed appropriately and rigorously? 

Reviewer #1: Yes

Reviewer #2: No

3. Have the authors made all data underlying the findings in their manuscript fully available?

Reviewer #1: Yes

Reviewer #2: Yes

4. Is the manuscript presented in an intelligible fashion and written in standard English?

Reviewer #1: No

Reviewer #2: No

5. Review Comments to the Author

Reviewer #1: The topic of the paper is intersecting. The abstract is clear in showing the objective of results of the study. Comments are as follows:

a. Figures should be started from Figure 1.

b. the solution of the figures should be improve and high quality figures are preferred.

c. there are too much content in section 2.

d. the paper should be reorganized as INTRODUCTION, DATA PREPARATION, METHODS, RESULTS, DISCUSSION, CONCLUSION

e. several Bayesian related studies could be acknowledged.

Real-time conflict-based Bayesian Tobit models for safety evaluation of signalized intersections. Accident Analysis & Prevention. 144, 105660.

A hierarchical Bayesian peak over threshold approach for conflict-based before-after safety evaluation of leading pedestrian intervals. Accident Analysis & Prevention, 147, 105772, DOI. 10.1016/j.aap.2020.105772

Reviewer #2: Reviewers’ Comments,

Owing to the rise in the popularity of automobiles over the last century, road accidents have become one of the leading causes of death in many countries around the world. The driver behaviors, such as speeding, drunk driving, and using a mobile phone while driving, are the major factors which lead to inattention of drivers. In addition to avoiding dangerous driving practices yourself, you must learn to identify unsafe behavior in other drivers. When you recognize the risk posed by such drivers, you can act pre-emptively to mitigate that risk by changing your speed, yielding the right-of-way or altering your position on the road. Hence, this manuscript seems an interesting piece of work, and the work sounds fine and good quality. The subject is interesting and emerging. However, I do suggest several important changes to address some weaknesses.

1. Authors are encouraged to add literature review section or related work, such as dangerous driving behavior, dangerous driving behavior identification methods, and so on. What’s more, what are the authors’ main contributions for the readers?

2. The literature review appears not completely updated. Some latest results on dangerous driving behavior, Bayesian approach and K-means algorithm can be found in:

Zhang, Q., Ge, Y., Qu, W., Zhang, K., Sun, X. (2018). The traffic climate in China: The mediating effect of traffic safety climate between personality and dangerous driving behavior. Accident Analysis & Prevention, 113, 213-223.

Guo, Y., Li, Z., Liu, P., Wu, Y. (2019). Modeling correlation and heterogeneity in crash rates by collision types using full Bayesian random parameters multivariate Tobit model. Accident Analysis & Prevention, 128, 164-174.

Guo, L., Zhou, J. B., Dong, S., Zhang, S. C. (2018). Analysis of urban road traffic accidents based on improved K-means algorithm. China J. Highway Trans, 31(4), 270-279.

Dong, S., Zhang, M., Li, Z. (2020). Risk analysis of vehicle rear-end collisions at intersections. Journal of advanced transportation, 2020, 1-11.

3. In the United States, the most common dangerous driving behaviors which result in collisions are: (a) speeding, (b) driving under the influence of alcohol, (c) distracted driving (including cell phone use and driving while fatigued), (d) reckless or aggressive driving (including tailgating, unsafe lane changes, running stop signs and failure to yield right-of-way), but in figure 2.13, why the authors only divide the driver's driving style into three types? In addition, how do the authors classify the dangerous driving behaviors? What is the detailed evidence for the classification? How is it different from other literatures? Please double check.

4. The authors employed Bayesian network to construct a vehicle-based traffic crash risk model. Please give more explains and detailed steps for the Bayesian network approach. It is suggested that the authors added the detailed process of the Bayesian network approach.

5. How to prevent these dangerous driving behaviors after the identification?

6. PLOS authors have the option to publish the peer review history of their article (what does this mean?). If published, this will include your full peer review and any attached files.

Reviewer #1: No

Reviewer #2: **Yes: **Dr. Zhou

---

## [Author Response · Author response to Decision Letter 0]

8 Apr 2021

Journal Requirements:

1.Please ensure that your manuscript meets PLOS ONE's style requirements, including those for file naming.

Response: Thank you very much for your reminder. We have modified the format of the article as required.

2.We suggest you thoroughly copyedit your manuscript for language usage, spelling, and grammar.

Response: We apologize for the language problems in the original manuscript. The language presentation was improved with careful review.

3.Funding information should not appear in the Acknowledgments section or other areas of your manuscript.

Response: Thank you very much for your reminder. We have deleted the funding information that appeared in the article.

4.Please amend either the abstract on the online submission form (via Edit Submission) or the abstract in the manuscript so that they are identical.

Response: Thank you very much for your reminder. We have amended the abstract on the online submission form.

5.Please ensure that you refer to Figure 2.2, 2.6, 2.7 and 4.4in your text as, if accepted, production will need this reference to link the reader to the figure.

Response: We are grateful for the suggestion. To be more clear and in accordance with the reviewer concerns, we have referred to each figure in our text.

6. We note you have included a table to which you do not refer in the text of your manuscript. Please ensure that you refer to Table 3.2 in your text; if accepted, production will need this reference to link the reader to the Table.

Response: We are grateful for the suggestion. To be more clear and in accordance with the reviewer concerns, we have referred to Table 3.2 in our text.

Reviewer #1

a. Figures should be started from Figure 1.

Response: We are sorry for the format error and we have corrected the name of tables and figures according to journal requirements.

b. the solution of the figures should be improve and high quality figures are preferred.

Response: Thank you for your suggestion. As suggested by reviewer, we deleted the unimportant pictures in the article, thereby simplifying the narrative of the article and at the same time, we have supplemented the conclusions of some pictures.

c. there are too much content in section 2.

Response: We are extremely grateful to reviewer for pointing out this problem. We have adjusted the structure of the article. In the second part of the article, only the data sources and data processing methods are introduced, and the results are displayed in the fourth part of the article. It is convenient for readers to understand and grasp the overall content of the article.

d. the paper should be reorganized as INTRODUCTION, DATA PREPARATION, METHODS, RESULTS, DISCUSSION, CONCLUSION.

Response: Thank you for your suggestion. As suggested by reviewer, we have re-adjusted the structure of the article and added the suggested content to the manuscript which make the structure of the article clearer and easier for readers to understand.

e. several Bayesian related studies could be acknowledged.

Response: We are grateful for the suggestion. As suggested by the reviewer, we have read the article about Bayesian research and added more details about Bayesian research in the first part and the third part of the article.

Reviewer #2

1. Authors are encouraged to add literature review section or related work, such as dangerous driving behavior, dangerous driving behavior identification methods, and so on. What’s more, what are the authors’ main contributions for the readers?

Response: We agree with the comment and have supplemented literature review section. As for authors’ main contributions, our reply is as follows: we developed 8 indicators to comprehensively identify dangerous driving behaviors. Compared with previous studies, this paper considered more types of dangerous driving behaviors, which can ensure the safety of vehicle operation better. And this paper proposes to use Bayesian network to build a vehicle operation risk assessment model which can not only predict the risk but can analyze the correlation between various factors in depth as well.

2. The literature review appears not completely updated. 

Response: We deeply appreciate the reviewer’s suggestion. According to the reviewer’s comment, we have supplemented the main methods and development process of Bayesian theory in the method part of the article and supplemented the background content in the introduction part of the article. And the cited documents have also been added and updated.

3. In the United States, the most common dangerous driving behaviors which result in collisions are: (a) speeding, (b) driving under the influence of alcohol, (c) distracted driving (including cell phone use and driving while fatigued), (d) reckless or aggressive driving (including tailgating, unsafe lane changes, running stop signs and failure to yield right-of-way), but in figure 2.13, why the authors only divide the driver's driving style into three types? In addition, how do the authors classify the dangerous driving behaviors? What is the detailed evidence for the classification? How is it different from other literatures? Please double check.

Response: Our deepest gratitude goes to you for your careful work. the discussion regarding this question is presented following: by reading relevant literature on dangerous driving behaviors in the past, we summarized 8 dangerous driving behaviors based on vehicle trajectory data which can ensure the safety of vehicle operation better. As for the classification of driving style, we use the K-means++ method to classify the driving style of the driver, and use the elbow method to determine the number of classifications. From Fig 4, we can see that when the number of clusters is less than 3, the change in SSE is large, and when the number of clusters is greater than 3, the change in SSE tends to be flat. Therefore, we determine that the number of classifications is 3.

4. The authors employed Bayesian network to construct a vehicle-based traffic crash risk model. Please give more explains and detailed steps for the Bayesian network approach. It is suggested that the authors added the detailed process of the Bayesian network approach.

Response: We deeply appreciate the reviewer’s suggestion. According to the reviewer’s comment, we have supplemented the main research process and results of Bayesian theory in the method part, and supplemented the details of the MCMC algorithm and counting algorithm used in the article. The specific steps and results of the algorithm are displayed in the results section.

5. How to prevent these dangerous driving behaviors after the identification?

Response: We are grateful for the suggestion. To be more clear and in accordance with the reviewer concerns, we have added a brief description as follows: once the risk level reaches a higher level, the model infers the most likely driving behavior that causes the danger based on the posterior probability, and reminds the driver to respond.

---

## [Decision Letter · Decision Letter 1]

17 May 2021

Examining Bayesian network modeling in identification of dangerous driving behavior

PONE-D-21-00035R1

Dear Dr. Cheng,

We’re pleased to inform you that your manuscript has been judged scientifically suitable for publication and will be formally accepted for publication once it meets all outstanding technical requirements.

Kind regards,

Yanyong Guo, Ph.D

Academic Editor

PLOS ONE

Additional Editor Comments (optional):

Reviewers' comments:

Reviewer's Responses to Questions

**Comments to the Author**

1. If the authors have adequately addressed your comments raised in a previous round of review and you feel that this manuscript is now acceptable for publication, you may indicate that here to bypass the “Comments to the Author” section, enter your conflict of interest statement in the “Confidential to Editor” section, and submit your "Accept" recommendation.

Reviewer #1: All comments have been addressed

Reviewer #2: All comments have been addressed

2. Is the manuscript technically sound, and do the data support the conclusions?

Reviewer #1: Yes

Reviewer #2: Yes

3. Has the statistical analysis been performed appropriately and rigorously? 

Reviewer #1: Yes

Reviewer #2: Yes

4. Have the authors made all data underlying the findings in their manuscript fully available?

Reviewer #1: Yes

Reviewer #2: Yes

5. Is the manuscript presented in an intelligible fashion and written in standard English?

Reviewer #1: Yes

Reviewer #2: (No Response)

6. Review Comments to the Author

Reviewer #1: (No Response)

Reviewer #2: Reviewer #1,

I appreciate the author's extensive revision work. All the comments have been addressed, the topic of this manuscript falls within the scope of PLOS ONE, the paper has improved significantly. I recommend it for publication.

Congratulations.

Thank you.

Your reviewer

7. PLOS authors have the option to publish the peer review history of their article (what does this mean?). If published, this will include your full peer review and any attached files.

Reviewer #1: No

Reviewer #2: No

---

## [Editor Report · Acceptance letter]

21 Jul 2021

PONE-D-21-00035R1 

Examining Bayesian network modeling in identification of dangerous driving behavior 

Dear Dr. Cheng:

I'm pleased to inform you that your manuscript has been deemed suitable for publication in PLOS ONE. Congratulations! Your manuscript is now with our production department. 

Kind regards, 

on behalf of

Dr. Yanyong Guo 

Academic Editor

PLOS ONE